# The Reciprocal Involvement of Family Business Owners and Communities in Business Success

**Jennifer Johnson Jorgensen [1,\*], Diane Masuo [2], Linda Manikowske [3] and Yoon Lee [4]**

[1] Textiles, Merchandising & Fashion Design, University of Nebraska-Lincoln, 205 Home Economics, Lincoln, NE 68583-0802, USA

[2] Family & Consumer Sciences, University of Hawaii at Manoa, Honolulu, HI 96816, USA; masuo@hawaii.edu

[3] Apparel, Merchandising, Interior Design, and Hospitality Management, North Dakota State University, Fargo, ND 58108, USA; linda.manikowske@ndsu.edu

[4] Human Development and Family Studies, Utah State University, Logan, UT 84322-2905, USA; yoon.lee@usu.edu

\* Correspondence: jbjorgensen@unl.edu; Tel.: +1-402-472-5462

**Abstract:** It is believed that highly involved business owners and community members will yield benefits to ensure business and community sustainability over time. However, little research has delved into understanding the role of business owners' involvement and the community's involvement in business outcomes. Thus, the purpose of this study was to investigate the reciprocal involvement of family business owners and the community. To investigate this phenomenon, this study utilized survey data from a rare group of business owners who currently operate long-standing businesses. Results indicate that more involved business owners perceived higher levels of business success. When seeking a profit, business owners also tended to be more involved in the community than owners not seeking a profit. However, family-owned businesses felt that the community did not contribute to their businesses and did not stay involved over time. Overall, business owners felt that they contributed more than the community provided in return. Recommendation is made to stress in entrepreneurship curricula the importance of reciprocal involvement between businesses and their communities and vice versa to promote business and community sustainability over time.

**Keywords:** community social responsibility; entrepreneurial interdependence; family-owned businesses; social capital

## 1. Introduction

According to the 2019 U.S. Small Business Administration (SBA), there are 30.7 million small businesses in the U.S., accounting for 99.9% of all businesses [1]. In the year 2016, small businesses in the U.S. employed 59.9 million people, which is 47.3% of the private workforce. Businesses with less than 100 employees make up the largest share of this business employment—many of which may be family businesses. Family-owned businesses are also key supporters of the local economy in terms of workers, capital, management, and entrepreneurial capabilities. Family-owned businesses not only provide income for their families, job opportunities for non-family members and bring dollars into the community [2], but they can also offer leadership and financial support for community projects, civic clubs, and other local organizations [3]. Family-owned businesses can foster local pride and contribute to the quality of life, making the community even more attractive for additional business opportunities and in-migration [4]. Understanding how family-owned businesses perceive success also helps to guide entrepreneurship education for future generations.

Family business owners have values which help guide business decisions. Not only do family-owned businesses focus on financial gain, but they also add to their stock of resources

by building social and emotional capital, which together help to sustain the business over time. Community social responsibility (CSR) also outlines businesses' commitment to the community, overall support provided to the community, and sense of community apparent in the local geographical area. Overall, family business owners perceive greater business success if they are more committed to the community in which the business operates [5]. In addition, varied perceptions of success can be explained by the amount of human and social capital present in a business [6].

As family businesses grow, more community support can be provided by the business [5]. Businesses that have been in the community longer are more likely to participate in community development [7], as social capital develops over time [8]. Further, as a business engages in socially responsible actions within a community, it is rewarded with support from customers, support staff, and stakeholders [9,10]. In relation to these findings, the relationships developed through a combination of social responsibility and social commitment lead to a competitive advantage for businesses [11]. Social capital has also been found to both initiate and prohibit actions between community members [8]. Thus, positive relationships between family businesses and the community help to build value and social capital [5].

This study examined owners' involvement in the community, which has been found to enhance the sustainability of small family-owned businesses in rural and small communities. Involvement in the community has been found to yield benefits to both communities and businesses, including financial performance. Social capital and financial capital are important for the sustainability of small family-owned businesses. While financial capital highlights the economic resources a business needs or has, social capital focuses on the relationships or social networks among people [12]. To develop social capital and financial capital, family business owners and the community should have a reciprocal relationship [5]. This reciprocal relationship has been associated with higher levels of perceived business success [13]; a well-educated, stable, satisfied workforce; a healthy business environment; and a thriving community in which to live and to do business [10,14,15]. In spite of these benefits, there has been limited research on how the level of involvement of businesses with their community and the reciprocal involvement of the community with the local business are related to business success among small family-owned businesses. The current study addresses this limitation by examining the level of involvement among a rare sample of long-standing, family-owned businesses.

Current entrepreneurial and community-based theories do not typically identify the level of involvement that both a business and community engage in during the business's life cycle. The Entrepreneurial Interdependence Theory, as found in Fortunato and Alter's [16] study, highlights the interconnectedness of individuals, institutions, and opportunities found within a bounded environment (e.g., municipality of a non-metropolitan county), also known as the community. In order for businesses to succeed, the community in which a business operates, through its institutions, must provide opportunities for start-up businesses. Institutions such as the local government, business groups and a network of businesses must contribute to a healthy business climate, and ultimately the individual business owners and institutions operating within the community must have shared interests [17]. Missing from this theory is the level of involvement between the three entities—individuals, institutions and opportunities.

This study utilized a rare sample of family business owners and operators that have been studied longitudinally over the past 19 years. Our sample represents the "survivors"—all of which are businesses that have survived for nearly two decades. The longevity of this sample allowed for the enhanced detection of patterns over time and the ability to determine factors contributing to long-term business success. Since these businesses are well established, this sample serves as an ideal case to determine patterns of involvement between the business owner and the community. It is also clear that social capital and CSR help to drive involvement, based on the review of literature. Thus, the purpose of this study is to understand how the level of owner involvement in the community (high involvement versus low involvement) and community involvement in local businesses (high involvement versus

low involvement) impact business success in long-standing family businesses. Additional owner and business characteristics were also investigated.

A series of research questions were posited:

(1) To what extent are the business owners and community involved in the community and in local businesses?
(2) What is the difference in business and owner characteristics among those who are highly involved in their communities and those who are less involved in their communities?
(3) What is the difference in business and owner characteristics among businesses located in communities that are highly involved and communities that have lower involvement with family businesses?
(4) Is there an association between the owner's level of involvement (high vs. low) in the community and business success?
(5) Is there an association between community level of involvement (high vs. low) in a family business and business success?

To answer our research questions, a series of statistical tests were run to determine the differences between business characteristics, owner characteristics, the varying level of involvement of both the owner and community, and business success. Our results support our adapted model presented in the literature review and highlight various characteristics that contribute to long-term business success. The results of this study can also be used to inform policy and programming for family businesses, community development, economic development, and business assistance for entrepreneurs and their communities. In addition, implications for entrepreneurship education are suggested based on the results of this study.

## 2. Literature Review

To provide a foundation for the study of the level of involvement among family businesses and communities, this review of literature includes five topics. The first topic is social capital, its definition, how it relates to community social responsibility, and the involvement that owners and communities contribute to each other. The second topic is community social responsibility and how businesses feel the responsibility to contribute to their communities. The third topic is family business involvement with local communities. The fourth topic is local community involvement with family businesses, and the last topic is the Entrepreneurial Interdependence Theory, which provides one of the theoretical foundations for this study.

### 2.1. Social Capital

Social capital is a critical community characteristic that can influence family businesses as well as be influenced by other types of capital, including human capital, political capital, financial capital, natural capital, cultural capital, built capital [18], and intellectual capital [8]. Social capital is an intangible resource that a family business establishes through the individual business owner's social networking relationships with individuals (customers, suppliers, competitors) and institutions (government and political entities, leaders of community, trade or employee organizations). This resource is more than the social network itself or the people linkages; it is embodied in information sharing and trust [19,20]. When members of the social network interact more, stronger social capital will result [8]. In order to succeed, a family business needs to utilize not only financial and human resources but also the intangible resource of social capital.

Social capital can be internal or external [12]. Internal social capital focuses on the social networking relationships among individual members within a system or organization. External social capital, which is within the scope of this research, is concerned with social networking relationships between an individual or organization and its relevant stakeholders (e.g., customers, suppliers, competitors, government officials, and leaders of community or trade associations). The social networking behaviors

can be observed through what is known as corporate/community social responsibility behaviors. The current study focuses on external social capital by examining how the level of owner involvement in the community and community involvement in the family business are associated with business success.

### 2.2. Community Social Responsibility

Social capital drives social responsibility within a community. Corporate social responsibility is defined as the sense of responsibility that businesses and their stakeholders have for each other's well-being and the environment that they share [5,21], while community social responsibility (CSR) focuses on social responsibility at the community level [5]. Researchers may vary in their definition of CSR, but there is agreement that CSR contributions by businesses go beyond providing goods and services to the marketplace. Family-owned businesses hold a unique perspective of socially responsible business behavior because of family ties to the community that they call home.

Niehm et al. [5] explored the antecedents and consequences of community social responsibility for family firms located in rural communities to determine whether their CSR orientation contributed to the performance of the business. The authors found that family-owned businesses identify three dimensions of CSR in their businesses and communities. The three dimensions included commitment to the community, community support, and sense of community. First, commitment to the community demonstrates the reciprocal relationship that communities and businesses have with one another. This dimension helps to create value for both the business and the community. Second, the community support dimension highlights how the small business gives back to the community in which it is based. Lastly, the sense of community dimension demonstrates the satisfaction and needs of the community from the lens of the small business owner.

Based on the results of Niehm et al.'s [5] study, family businesses deemed their commitment to the community as the most important dimension of CSR. The commitment to the community dimension was also significantly related to subjective business performance. This means that the more committed the family business was to the community, the more successful the business perceived itself to be. In addition, a greater level of community support yields a greater objective performance for the business [5]. Thus, CSR serves as a motivating factor for the involvement that owners and communities contribute to one another.

### 2.3. Family Business Involvement with Local Communities

Family firms' commitments to the community, as well as community involvement, were associated with higher levels of business success [13]. Some of the success measures included a well-educated, stable, satisfied workforce; healthy business environment; and thriving community in which to live and do business [10,14,15]. Business involvement in a community can also enhance the firm's public image and prestige, which can lead to increased sales and the offering of favorable loan rates from bank loan officers. Besser and Miller [22] identified owner, business and community characteristics associated with increases and decreases in CSR activity. However, there are no known studies that have extended Besser and Miller's [22] research by examining how the level of involvement of owners and the community in which the business operates influence business success among small family-owned businesses.

Business owners that were not satisfied with a community tended to support future community development [7]. However, a strong sense of community did not translate to how family business owners supported the community [5]. If a community issue arises, small businesses determine whether a community's claim toward the topic is appropriate and/or critical before taking action. Overall, appropriate and critical calls to action from the community are perceived to be more important than the community using its own power to enact change. Small businesses were aware of the reciprocal relationship that they had with the community, and engaging in CSR initiatives helped to demonstrate responsibility in the local community [23].

### 2.4. Local Community Involvement with Family Businesses

Local communities can provide various types of support, as Barraket, Eversole, Luke, and Barth [24] determined that small business owners accessed a myriad of philanthropic resources and/or governmental funding sources at some point during the business's lifespan. Small businesses in rural areas were more likely to access governmental funding sources that would further support the local community's development goals. Rural businesses were also more likely to rely on local community fundraising efforts and donations from community members. Despite overall community support, support from other local businesses was scarce [24].

Community support and the power that the community has to make changes can be divided. As found by Park and Campbell [23], local communities did not have the power to drive concern among small businesses toward CSR initiatives. Interestingly, local communities tended to remain indifferent until an urgent issue unfolded. The urgent issue also motivated small businesses and community members to band together to take action [23]. More established businesses were also able to use internal resources (i.e., new ventures, new products) to benefit the communities around them. These new ventures and products helped build social capital and financial capital. For example, a business may provide work opportunities to underserved populations or highlight the reusability of a product [24]. Despite the challenges that communities may have when working with local businesses, the interactions between each entity can be better understood through the Sustainable Family Business Theory and the Entrepreneurial Interdependence Theory.

### 2.5. Theoretical Frameworks

Small family businesses are not always focused on financial success but may be more focused on the sustainability of the business over time. The Sustainable Family Business Theory (SFBT) outlines how the sustainability of a business relies on a combination of the business's financial success and family functionality, including family and business resources and constraints, disruptions in family or business transactions, and family and firm structure [6,25]. The SFBT was utilized in this study to outline how family-owned businesses operate within a community during varied times, which may lead to business success. Thus, businesses depend on the interdependent relationships among various entities to ensure a successful business.

Businesses and communities do not operate independent of one another. In a study investigating entrepreneurial behavior, Fortunato and Alter [16] highlighted the interdependencies that businesses required to be successful as outlined by the Entrepreneurial Interdependence Theory. They argued that interdependent relationships among the individual entrepreneur, the institutions surrounding the business, and the opportunities available to start a business were necessary to form a successful business model. While individual motivations and opportunities to start a business have been the focus of many research studies, some studies have also investigated institutional influences and the interactions among these three entities [16,17]. In past studies, institutions have been broadly explained as formal or informal relationships that enable and constrain opportunities [17]. For this study, the institution is defined as each business's support system of local businesses and organizations within a community. Opportunities to start a business was not investigated for this study, as the current sample includes 71 businesses that have survived for over 19 years. Thus, to further build upon Fortunato and Alter's [16] model, the researchers of this study sought to understand how the level of owner involvement in the community (high involvement versus low involvement) and community involvement in local businesses (high involvement versus low involvement) impact business success for existing family-owned businesses. This study's model, adapted from Fortunato and Alter's [16] study, is available in Figure 1.

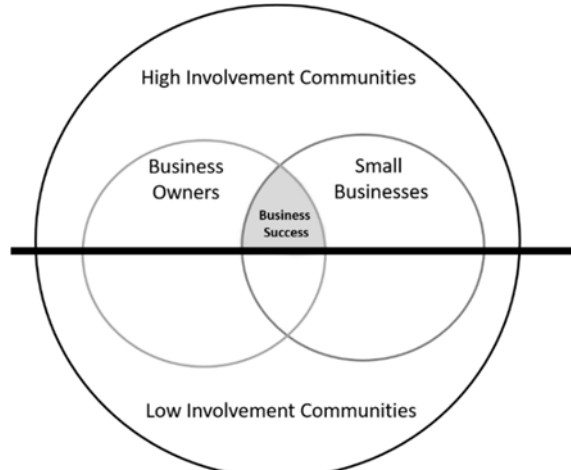

**Figure 1.** Adapted Entrepreneurial Interdependence Theory Focusing on Community Involvement.

The adapted model, based on the Entrepreneurial Interdependence Theory, posits that individual business owners and other small businesses and organizations operating in high-involvement communities will perceive greater business success. Business success may include financial success, general business success, or business goal achievement. Small businesses in low involvement communities will not perceive the same business success as compared to those in high involvement communities. Overall, the level of involvement is believed to be driven by social capital and CSR. Based on this theory and literature, the following hypotheses have been posited:

**Hypothesis 1.** *Long-standing family-owned businesses will have highly involved owners and communities.*

**Hypothesis 2.** *Long-standing family-owned businesses that have highly involved owners and highly involved communities will perceive a greater level of business success.*

Thus, this study's overarching aim is to understand the role of involvement in family-owned businesses and the communities that they serve.

## 3. Materials and Methods

### 3.1. Data and Sample

This study utilized a rare, longitudinal sample of 71 family-owned businesses that were surveyed over the course of 19 years. Data for the longitudinal study was collected from a national family business sample in the years 1997, 2000, 2007, and 2016. The National Family Business Study (NFBS) began with Wave 1 in 1997, with 708 family firms with complete information from a household manager and a business manager within each household [26]. Due to attrition, by Wave 2 in 2000, this decreased to complete household manager and business manager data from only 529 family firms [27]. By 2007, Wave 3, 128 firms closed, leaving 395 firms that were still in business. However, six firms did not report the starting year of their business, so these cases were dropped, leaving complete information from only 523 family firms [28].

After a nine-year hiatus, Wave 4 was launched in 2016. The initial sample consisted of 326 firms, but only 180 of the firms were still in business. Of this number, only 149 could be verified or screened; 58 of the firms closed, were no longer owned or had a different manager from the one interviewed in 2007. Of the remaining 91 firms, 18 refused to participate or were dropped after the maximum telephone call-backs were reached. The final sample in 2016 consisted of 73 firms that were still open and owned and managed by the same person interviewed in Wave 3. Of the 73 firms, two were deleted

because of incomplete information, leaving a sample of 71 firms. A distinct focus of the 2016 sample is that all firms were located in communities with populations of less than 10,000 [29].

### 3.2. Data Collection Procedure

A survey research center at a large Midwestern university was contracted to conduct semi-structured telephone interviews. The interviews began with a 5-min screener, followed by a 27-min full interview. The interview instrument included items from the previous 1997, 2000, and 2007 waves for continuity and to enable tracking over time. Additional questions included firm performance, managerial decision-making processes or practices, online business strategies, and the potential impact of the 2007 recession on the firm. Several open-ended items were included to accommodate variations in business and household situations. The data utilized in this study is proprietary and can be provided upon request.

### 3.3. Sample Characteristics

A unique sample of family business owners that have been studied longitudinally over the past 19 years was analyzed for this study. Data analyses were completed with 71 family-owned businesses that were still open in 2016. To answer the main research questions, means, frequencies, percentages, and cross-tab analyses were performed. Of the 71 businesses that survived over time, a majority of owners (73%) were between 50 and 69 years old, male (65%), and over half (55%) had earned a B.A. degree. The majority of these businesses had three or fewer employees (52%) and were predominantly sole proprietorships (38%).

### 3.4. Variables

#### 3.4.1. Dependent Variable

In this study, the business success of a family-owned business was the main dependent variable. Business success was measured by three variables—make a profit, owners' perceived business success, and business owners' goal achievement. For the profit variable, if owners answered "make a profit," it was coded as 1, and if they answered "lose money" or "break even," these responses were coded as zero. In the survey, business owners were asked "Overall, how successful is your business now?" The responses to this question ranged from 1 (not at all successful) to 5 (extremely successful). If they answered this question as "4 = successful" or "5 = very successful," it was coded as 1, and if they answered this question as "1 = not at all successful, 2 = not successful, or 3 = neutral," these responses were coded as zero. As for the goal achievement variable, business owners were asked whether they agreed with "I am accomplishing what I wanted to do with my business operation." The responses to this question ranged from 1 (strongly disagree) to 5 (strongly agree). A value of 1 was assigned to those who answered "4 = agree" or "5 = strongly agree" and a value of zero was assigned to those who answered "1 = strongly disagree, 2 = disagree, or 3 = neutral."

#### 3.4.2. Independent Variables

To measure the level of involvement between family businesses and communities, two types of involvement variables were created. For the owners' involvement in community variable, CS6, CS7, CS9, and CS8 in Table 1 were utilized, whereas, for the community involvement in local business variable, CS2, CS3, CS 4, and CS5 in Table 1 were used for analyses. Business owners responded to the CS2–CS9 questions "from 1 = strongly disagree, 2 = disagree, 3 = neutral, 4 = agree, 5=strongly agree."

Other explanatory variables included in the cross-tab analyses were owner and business characteristics. Owner characteristics included gender (males, females), age (<65, 65+), education level (high school graduates, some college, college degree), health status (fair/good, excellent), and their views on business either as a way of life or as a means to earn revenue. Business characteristics

included business size (zero employees other than owner, 1–2 employees, 3+ employees) and region (rural—less than 2500 in population, town—2500–10,000 in population, city—10,000+ in population).

**Table 1.** Extent of Business Owner and Community Involvement (N = 71).

| | Business Owner Involvement in Community | | | | Community Involvement in Local Businesses | | |
| --- | --- | --- | --- | --- | --- | --- | --- |
| | | Mean [1] | % [2] | | | Mean [3] | % [4] |
| CS6 | I help other businesses in the community | 3.8 | 71.8% | CS5 | People in the community helping my business | 3.3 | 47.9% |
| CS7 | My business participates in organizations in the community | 2.8 | 26.8% | CS3 | Community organizations stay involved in my business | 2.5 | 18.3% |
| CS9 | I hold positions on community boards and work to improve the community | 2.9 | 38.0% | CS4 | Community boards make decisions that benefit my business | 2.5 | 18.3% |
| CS8 | My business donates to and helps important programs in the community | 3.7 | 67.6% | CS2 | The community supports local businesses through various networks | 3.1 | 35.2% |

Note: [1] Mean—represents a mean number on response, ranging from 1 = strongly disagree, 2 = disagree, 3 = neutral, 4 = agree, to 5 = strongly agree; [2] %—represents the percentage of those who answered agree or strongly agree, indicating that those business owners who are highly involved in the community; [3] mean—represents a mean number on response, ranging from 1 = strongly disagree, 2 = disagree, 3 = neutral, 4 = agree, to 5 = strongly agree 1 = strongly disagree, 5 = strongly agree; [4] %—represents the percentage of those who answered agree or strongly agree, indicating that the community is highly involved in local business support.

## 4. Results

Table 1 presents descriptive results of the extent of business owner and community involvement within the community and in local businesses. Among the four categories of business owner involvement in the community, the mean value of owners' community involvement for CS6—"I help other businesses in the community"—was the highest (M = 3.8), showing that 71.8% of business owners reported that they helped other businesses in the community. The mean value of CS8—"My business donates to and helps important programs in the community"—was also higher than the other items. Approximately 68% of business owners reported that their businesses donated and helped important programs in the community. On the other hand, the mean value of perceived community involvement for CS7—"My business participates in organizations in the community"—was the lowest (M = 2.8), showing that only 26.8% of the study sample reported that their businesses participated in an organization in the community.

Regarding the community's involvement in local businesses, the mean value of owners' perception of community involvement in local businesses for CS5—"People in the community helping my business"—was the highest (M = 3.3), but the mean values of both CS3 (community organizations stay involved in my business) and CS4 (community boards make decisions that benefit my business) were the lowest (M = 2.5). These two categories also explain that only 18.3% of the study sample felt that community organizations were not closely working with local businesses, and community boards' decisions were not beneficial to their businesses.

According to Table 1, it seems that in all categories, business owners perceived that they contributed more to the community than the community contributed to their businesses. Most often, business owners felt that they had helped other businesses in their community and supported local community programs. However, they were less likely to participate in organizations within the community or be community board members. Less than half of the business owners (35.2%) perceived that they were being supported by the community. While 47.9% of the sample believed that people in the community helped their business, 18.3% believed that community organizations and boards did little to benefit their business.

Table 2 presents four types of business owner and community involvement. The involvement-type variables were created using the information from the "Individual Owner Involvement" categories (CS 6, CS 7, CS 8, and CS 9). The "Community Involvement" categories (CS 2, CS 3, CS 4, and CS 5) were also used. If the business owners responded to each of these questions as "agree" or "strongly agree," a value of 1 was assigned (high involvement). Likewise, if business owners responded to these items as "neutral," "disagree," or "strongly disagree," a value of zero was assigned (low involvement). Each of the CS series variables was totaled, and one variable was created that represents business owner involvement in the community (1 = high involvement, 0 = low involvement) as well as community involvement in the local business (1 = high involvement, 0 = low involvement).

**Table 2.** Four Types of Business Owner and Community Involvement (N = 71).

| Involvement Type | Business Owner Involvement in Community | Community Involvement in Local Business | N | % |
|---|---|---|---|---|
| Type 1—**LL** | **Low** Involvement [1] | **Low** Involvement | 9 | 12.7% |
| Type 2—**HL** | **High** Involvement [2] | **Low** Involvement | 18 | 25.3% |
| Type 3—**LH** [3] | **Low** Involvement | **High** Involvement | 3 | 4.2% |
| Type 4—**HH** | **High** Involvement | **High** Involvement | 41 | 57.8% |

Note: [1] Low Involvement—those who responded strongly disagree, disagree, and neutral; [2] High Involvement—those who responded agree and strongly agree; [3] LH represents those who responded low involvement in community and high involvement in local business.

Based on this analysis, Type 1—LL (Low, Low) means that both the business owners and their communities were less involved with one another. A category of Type 2—HL (High, Low) means that while business owners were highly involved in the community, the communities were less involved in their local businesses. The category of Type 3—LH (Low, High) displays that business owners were less involved in the community, whereas their communities were highly involved in local businesses. Lastly, Type 4—HH (High, High) represents the group of family-owned businesses, where business owners believed that they contributed a lot to the community, and they also received a lot from the community in return.

Table 2 shows that more than half of business owners (57.8%) fall into the Type 4 category, in which both owners and communities gave and received support within the community. However, approximately one-quarter of business owners (25.3%) believed that their businesses were highly involved in the community, but they received weaker support from the community. It can also be determined that 12.7% of business owners felt that they were not very involved in their community, and they felt their communities did little to support them. A few business owners (4.2%) reported that they were less involved in the community, but they had greater community involvement in local businesses. From Table 2, it was found that out of 71 survived family-owned businesses over the past 19 years, more than half of these business owners were highly involved in the community, and they also perceived that they had stronger communities. Therefore, Hypothesis 1 was supported.

In this study, differences in business and owner characteristics between those who were highly involved in their communities and those who were less involved in their communities were investigated. The difference in business and owner characteristics between businesses with stronger communities and businesses with weaker communities was also explored. Because the sample size is only 71 family-owned businesses, most of the test statistics were not statistically significant. Table 3 shows that certain groups were more likely to be highly involved in their community. These groups included males, owners under age 65, owners with college degrees, owners in excellent health, owners' working to earn money, businesses with 1–2 employees, or those located in a small town.

Table 3 shows that while female owners were less likely to be involved in their community than male owners, more female owners felt supported by their community than male owners. Owners less than 65 years old believed that they were highly involved in their communities compared to older business owners, and they also felt that their community was strong. This perspective was similar for

owners with a college degree, suggesting that highly educated business owners were more involved in their communities, and they felt their businesses had stronger community involvement as compared to owners with a high school diploma or those with some college education.

**Table 3.** Owner and Business Characteristics by Involvement Type (N = 71).

| Variables | Business Owner Involvement in Community | | | Community Involvement in Local Business | | |
|---|---|---|---|---|---|---|
| | Low (n = 12) | High (n = 59) | Test Statistics | Low (n = 27) | High (n = 44) | Test Statistics |
| Owner Characteristics | | | | | | |
| Gender: | | | | | | |
| Male | 50.0% | **67.8%** | $\chi^2 = 1.38$ | 66.7% | 63.6% | $\chi^2 = 0.07$ |
| Female | 50.0% | 32.2% | | 33.3% | **36.4%** | |
| Age: | | | | | | |
| < 65 | 41.7% | **64.4%** | $\chi^2 = 2.16$ | 59.3% | **61.4%** | $\chi^2 = 0.03$ |
| 65+ | 58.3% | 35.6% | | 40.7% | 38.6% | |
| Education: | | | | | | |
| High school | 25.0% | 20.3% | | 22.2% | 20.5% | |
| Some college | 41.7% | 30.5% | $\chi^2 = 1.02$ | 33.3% | 31.8% | $\chi^2 = 0.08$ |
| College degree | 33.3% | **49.2%** | | 44.5% | **47.9%** | |
| Health: | | | | | | |
| Fair/Good | 83.3% | 61.0% | $\chi^2 = 2.18$ | 74.1% | 59.1% | $\chi^2 = 1.65$ |
| Excellent | 16.7% | **39.0%** | | 25.9% | **40.9%** | |
| Way of Life: | | | | | | |
| B way of life | 83.3% | 66.1% | $\chi^2 = 1.38$ | 85.2% | 59.1% | $\chi^2 = 5.33$ ** |
| B earn money | 16.7% | **33.9%** | | 14.8% | **40.9%** | |
| Business Characteristics | | | | | | |
| Business Size: | | | | | | |
| No employee | 41.7% | 40.7% | | 44.4% | 38.6% | |
| 1–2 employees | 25.0% | **28.8%** | $\chi^2 = 0.08$ | 18.5% | **34.1%** | $\chi^2 = 2.09$ |
| 3+ employees | 33.3% | 30.5% | | 37.1% | 27.3% | |
| Region: | | | | | | |
| Rural | 33.3% | 19.7% | | 25.9% | 25.0% | |
| Town | 25.0% | **31.0%** | $\chi^2 = 0.81$ | 33.3% | **36.4%** | $\chi^2 = 0.97$ |
| City | 41.7% | 33.3% | | 40.8% | 38.6% | |

Note: * $p < 0.10$, ** $p < 0.05$, and *** $p < 0.01$.

Table 3 shows that those with excellent health were more involved in their community and received higher community support compared to those with fair/good health. Business owners sometimes run the business as their way of life or to make money as an income. Those who viewed their business as a source of income were more likely to be involved in their community. Similarly, the business owners who ran their business to earn money felt more community support. It is interesting to note that when businesses had 1 or 2 employees, they were more involved in their community and felt greater community support. When businesses were located in rural or city communities, they were less involved in their community and did not feel strong community involvement. However, businesses owners operating businesses in small towns (2500–10,000 in population) had strong involvement in the community and viewed their communities to be strong.

Table 4 shows the association between the four types of business owner/community involvement and business success in small family-owned businesses. One important research question of the current study was to examine how the business owners' involvement in their communities is associated with business success (business profit/perceived success/goal achievement) among small family-owned businesses. Another important question was to examine whether there is an association between the owner's high involvement in the community and business success (business profit/perceived success/goal achievement). Table 4 shows the results of cross-tab analyses.

**Table 4.** The Association between Four Types of Business Owner/Community Involvement and Business Success in Small Family-Owned Businesses (N = 71).

| Involvement Type | Make a Profit | | Perceived Business Success | | Business Goal Achievement | |
|---|---|---|---|---|---|---|
| | No (n = 26) | Yes (n = 45) | No (n = 12) | Yes (n = 59) | No (n = 19) | Yes (n = 52) |
| Type 1—**LL** | 8.5% | 4.2% | 4.2% | 8.5% | 4.2% | 8.5% |
| Type 2—**HL** | 5.6% | 19.7% | 4.2% | 21.1% | 5.7% | 19.7% |
| Type 3—**LH** | 0.0% | 4.2% | 0.0% | 4.2% | 0.0% | 4.2% |
| Type 4—**HH** | 22.5% | 35.3% | 8.5% | 49.3% | 16.9% | 40.8% |
| | $\chi^2 = 6.94$ * | | $\chi^2 = 2.49$ | | $\chi^2 = 1.61$ | |

Note: * $p < 0.10$, ** $p < 0.05$, and *** $p < 0.01$.

The profit variable responses were coded as "make a profit" =1; and "lose money" or "break even" = 0. The perceived business success variable was coded as 0 for response values from 1 to 3, and 1 for values 4, "successful" or 5, "very successful" variable, the responses of "lose money," "break even," and "make a profit" were assigned values "make a profit" = 1; "lose money" or "break even" = 0. For the perceived business variables, if business owners answered "4 = successful or 5 = very successful", it was coded as 1, and 0 if otherwise. For business goal achievement, if business owners answered "4 = agree or 5 = strongly agree", it was coded as 1, and 0 if otherwise.

A larger proportion of the HH group (Type 4) reported that they perceived their businesses to be successful (49.3%), and these owners were more likely to achieve their business goals (40.8%). In contrast, a larger proportion (8.5%) of the LL group (Type 1) reported that they did not make a profit through their businesses. When the business owners were highly involved in the community, but they felt that they had weak community support, this HL group (Type 2) showed a lower percentage (21.1%) of business success compared to the HH group (Type 4) (49.3%). The results in Table 4 show that when business owners' involvement in the community was high (e.g., Type 2 or Type 4), these owners were more likely to feel successful than those who were less involved in their community (e.g., Type 3). Not only did the HH group (Type 4) make a profit more often, but they also perceived their business to be successful and felt that they had reached their business goals. The LL group (Type 1), those who did little in their community and received little support in return, were less likely to profit, feel successful, or reach their business goals. Therefore, Hypothesis 2 was supported.

## 5. Discussion

Novel to this study, a rare sample of 71 long-standing businesses were utilized to determine patterns of involvement between family businesses and the communities in which they operate, and how the level of involvement is associated with business success. The results of this study indicate that the more a family business is involved in the community, the greater perceived success the business has achieved. Similarly, Besser [13] determined that community involvement is associated with higher levels of business success. In contrast, Haynes et al. [30] noted that owners' subjective perception about the community was more important in determining the success of the firm than owners' involvement (participating in business, social and other organizations) and participation in the community. To overcome this discrepancy, it is important that future research on this topic include more specific metrics for both involvement and business success and continue to adapt the contribution of involvement in the Entrepreneurial Interdependence Theory.

The Entrepreneurial Interdependence Theory was adapted in this study to include the variable of involvement. Fortunato and Alter [16] found that successful businesses needed to have interdependent relationships among entrepreneurs, local institutions, and the overall community. Since our sample focused on long-standing family businesses, it became clear that that success is tied to the level of involvement found among family business owners and other businesses and organizations in the community. Thus, this adapted theory should further guide additional research on the topic of the impact of owner and community involvement on business success.

Involvement in the community helps to build social capital and CSR. However, family business owners believed that they contributed more to the community than the community provided in return and that the community does not make positive decisions that benefit the family business. It is also perceived that community organizations do not stay involved in family businesses over time. In support of our results, Park and Campbell [23] determined that communities tend to be distant until an urgent issue is presented which requires united actions. It is unclear, however, whether businesses choose to disengage with the community when a community need is not present. These results further solidify that little is known on how community involvement impacts family business success over time.

Family business owners in this study stated that they were willing to help other businesses in their community. In contrast to our results, Barraket et al. [24] found that support from local businesses is scarce even when there is perceived community support. The difference may be between a business's intentions and actual behaviors toward other businesses in the community. This study also demonstrated that family business owners were also limited in their participation in community organizations, which may limit linkages between businesses in the same community. However, family businesses driven to make a profit were found to be more involved in their communities than those who are more interested in other forms of business success. Thus, it is important that the role of community involvement on business success is further researched and discussed by business and entrepreneurship professionals.

### 5.1. Limitations

This study has a few limitations that must be considered. First, this study has a limited sample size of 71 participants. However, this sample is unique, as these business owners have been studied over the course of 19 years. As such, the research team deems the sample to be valuable, as longitudinal patterns can be detected over time. Second, our sample of business owners identified the level of community involvement from their own perspective. Thus, business owners may be missing or overlooking some elements of community involvement present in their local area that could impact overall business success. Third, the measure of business success consists of three different variables, including perceived success, making a profit, and goal achievement. However, the research team would argue that people determine success based on a variety of reasons that may not be strictly economic.

### 5.2. Implications

Since business success can be traced back to the level of owner involvement in the community, programming and business assistance for family businesses should encourage community and network growth. The results of this study can also help to inform policy on business and economic development within the community. Failed businesses can have a great impact on the future viability of the community and the livelihood of their residents.

The results of this study can also be of value to entrepreneurship education programs in developing a curriculum that meets the needs of the growing number of young entrepreneurs. It is important to identify the entrepreneurial characteristics and strategies that increase the opportunities for family businesses to succeed. Thus, an entrepreneurship curriculum should include ways to build social capital, CSR, and community involvement among future business owners.

### 5.3. Directions for Future Research

The current study contributes to previous literature in two primary ways. First, our results indicated that the adaptation of the Entrepreneurial Interdependence Model was accurate in outlining the overlap between business owner involvement and community involvement on business success. To solidify the adaptation of the Entrepreneurial Interdependence Model, additional research should include other possible barriers toward business success that can exist in a bound environment. To examine the impact that level of community and business involvement have on business demise, future studies should include businesses that closed.

Second, our study has investigated the level of community involvement and its impact on business success. In this study, community involvement was reported by the business owner. Future studies should include a measure of community involvement from the community's perspective to increase understanding of the reciprocal relationship between level of involvement and business success. Obtaining a larger sample size would enable researchers to identify the level of involvement by industry classification of the business.

## 6. Conclusions

The results from this study suggest that when family businesses are more involved in their communities, the communities tend to reciprocate with higher levels of involvement in the family businesses. Interestingly, these higher levels of involvement were associated with higher levels of perceived business success by the owners. This relationship reinforces the importance for family businesses of building their social capital and CSR through involvement with their communities. The interdependence among various community entities also highlights the need for involvement to be included in future iterations of the Entrepreneurial Interdependence Theory, which may help prospective and current businesses develop more successful business models.

**Author Contributions:** Conceptualization, J.J.J., D.M., L.M. and Y.L.; methodology, J.J.J., D.M., L.M., and Y.L.; data analysis, Y.L.; writing—original draft preparation, J.J.J., D.M., L.M., and Y.L.; writing—review and editing, J.J.J., D.M., L.M., and Y.L. All authors have read and agreed to the published version of the manuscript.

**Funding:** This research received no external funding.

**Acknowledgments:** A special thank you to Linda Niehm, Margaret Fitzgerald, Glenn Muske, and Eonyou Shin, as well as members of the NC1030 research group, for their suggestions and support for this research.

**Conflicts of Interest:** The authors declare no conflict of interest.

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
