# Peer review of "The Reciprocal Involvement of Family Business Owners and Communities in Business Success"

_sustainability, doi:10.3390/su12104048_

Round 1

Reviewer 1 Report

Introduction

Line 94-101: Research questions would be more visible if it were placed each in a row.

I suggest that the rest of the paper be briefly presented at the end of the introduction section.

Literature review

Line 224: The source of figure 1 should be provided under the figure

Materials and Methods

Line 261-263: It is said that: “The survey center interviewed as many family business owners and managers as possible whose businesses were still open.” Does that mean that not all the family business owners whose businesses were still open were interviewed? If not, how were selected the 71? Moreover, how was selected the original sample? Considering the answers to the above, please make comments on the representativeness of the sample and (consequently) the relevance of the study.

The results suggest that within the paper, only the answers from the 2016 wave was analyzed (N = 71). If this is the case, that should be made clear within the paper. Moreover, when talking about the other three waves, for those information references should be provided. The same suggestion for the research instrument.

Line 55-56: It is said that “In addition, all firms were located in rural communities with populations less than 10,000.”, and later on (line 300-302): “Business characteristics included … and region (rural—less than 2,500 in population, town--2,500-10,000 in population, city--10,000+ in population)”. Please make the corrections where needed.

Results

Table 3: Besides the χ2 value, the significance level (p-value) should be provided. The readers should have the information to make their judgments. Moreover, it should be specified what statistic test was performed.

Table 3: Please correct the table so few times to be placed next to appropriate characteristics.

Conclusions

The conclusions section should be outlined the study implications, limitations, and future research directions. Among the limitations, the small sample size should be included.

Reviewer 2 Report

The paper is interesting. However, I suggest to improve the theoretical part of paper. Some issues have been mentioned in the lines 107-112 (social capital, family business involvement with local communities, local communities involvement with family business, entrepreneurial interdependence theory) but in the literature review I also see the section about CSR (based on two articles not new ones), and a paragraph about the sustainable family business theory. Do the section about social capital is really needed in this paper? I suggest to focus on significant topics related to the paper.

I appreciate the longitudinally studies. In the section materials and methods data, sample and variables are clearly presented but lack is of method used in the researches.

Hypothesis testing requires a clear indication of which hypotheses have been confirmed and which have been rejected.

I suggest you to emphasize more your opinion in conclusion.

Reviewer 3 Report

It is my pleasure to review the paper. While the paper is well-written and the data is strong, I have the following concerns:

  1. The main concern I have with the paper is the survival bias of the sample. The 71 businesses selected over 22 years are all survivors. Thus, the failed businesses are not counted in the sample.
  2. The measurement of business success is also problematic. It was measured by three variables -make a profit, owners’ perceived business success and business owners’ goal achievement. “Perceived success” and “Goal achievement” are subjective rather than objective. Even the “make a profit” variable is not reliable since most entrepreneurs are reluctant to accept failure and admit that they lose money.
  3. The measurement of involvement is on one side only (the small business owner side) and common method bias could be an issue if you don’t explain how do you work with the panel data. It will be more reliable if you can also provide the perception of involvement on the community side.
  4. You didn’t control the industry of the businesses and the nature of each business could have a different degree of involvement with the community. For example, a grass mowing company may have higher involvement with the community while the heater installation company has lower involvement.
  5. Most of the results are obvious. It adds little to our understanding of the relationship between small businesses and the community.

Round 2

Reviewer 1 Report

I appreciate the changes made by the aurhors

Reviewer 2 Report

I have no comments

Reviewer 3 Report

The manuscript has been improved dramatically in last round of R&R. However, it looks like most of my concerns have been moved to the directions of future research. I have no other questions.